# Targeted Two-Step Delivery of Oncotheranostic Nano-PLGA for HER2-Positive Tumor Imaging and Therapy In Vivo: Improved Effectiveness Compared to One-Step Strategy

**DOI:** 10.3390/pharmaceutics15030833

**Published:** 2023-03-03

**Authors:** Victoria O. Shipunova, Elena N. Komedchikova, Polina A. Kotelnikova, Maxim P. Nikitin, Sergey M. Deyev

**Affiliations:** 1Moscow Institute of Physics and Technology, Dolgoprudny 141701, Russia; 2Shemyakin–Ovchinnikov Institute of Bioorganic Chemistry, Russian Academy of Sciences, Moscow 117997, Russia; 3Nanobiomedicine Division, Sirius University of Science and Technology, Sochi 354340, Russia

**Keywords:** breast cancer, polymer nanoparticles, membrane-associated receptors, oncotheranostics, Nile Blue, doxorubicin, barnase, DARPin, personalized medicine

## Abstract

Therapy for aggressive metastatic breast cancer remains a great challenge for modern biomedicine. Biocompatible polymer nanoparticles have been successfully used in clinic and are seen as a potential solution. Specifically, researchers are exploring the development of chemotherapeutic nanoagents targeting the membrane-associated receptors of cancer cells, such as HER2. However, there are no targeting nanomedications that have been approved for human cancer therapy. Novel strategies are being developed to alter the architecture of agents and optimize their systemic administration. Here, we describe a combination of these approaches, namely, the design of a targeted polymer nanocarrier and a method for its systemic delivery to the tumor site. Namely, PLGA nanocapsules loaded with a diagnostic dye, Nile Blue, and a chemotherapeutic compound, doxorubicin, are used for two-step targeted delivery using the concept of tumor pre-targeting through the barnase/barstar protein “bacterial superglue”. The first pre-targeting component consists of an anti-HER2 scaffold protein, DARPin9_29 fused with barstar, Bs-DARPin9_29, and the second component comprises chemotherapeutic PLGA nanocapsules conjugated to barnase, PLGA-Bn. The efficacy of this system was evaluated in vivo. To this aim, we developed an immunocompetent BALB/c mouse tumor model with a stable expression of human HER2 oncomarkers to test the potential of two-step delivery of oncotheranostic nano-PLGA. In vitro and ex vivo studies confirmed HER2 receptor stable expression in the tumor, making it a feasible tool for HER2-targeted drug evaluation. We demonstrated that two-step delivery was more effective than one-step delivery for both imaging and tumor therapy: two-step delivery had higher imaging capabilities than one-step and a tumor growth inhibition of 94.9% in comparison to 68.4% for the one-step strategy. The barnase*barstar protein pair has been proven to possess excellent biocompatibility, as evidenced by the successful completion of biosafety tests assessing immunogenicity and hemotoxicity. This renders the protein pair a highly versatile tool for pre-targeting tumors with various molecular profiles, thereby enabling the development of personalized medicine.

## 1. Introduction

Biocompatible and biodegradable nanostructures offer great opportunities in the diagnostics and treatment of a wide range of diseases, including cancer [1,2,3,4]. Several dozen drugs based on nanostructures (protein polymer, iron oxide nanoparticles, liposomes) are already used in clinical practice [5,6]. In particular, more than twenty drugs based on a copolymer of lactic and glycolic acids, PLGA, are currently used to treat diseases such as breast and prostate cancer, schizophrenia, eye diseases, and others [5]. The success of PLGA-based nanomedications is primarily caused by their biocompatibility, as well as the ability to encapsulate a wide variety of both hydrophilic and hydrophobic substances.

Considering the anticancer properties of medical nanostructures, it is worth noting that they reach the tumor due to the effect of increased permeability and the retention of tumor vessels (EPR effect), which allows the accumulation of a significant proportion of injected nanoparticles in the tumor site. However, recent studies revealed that the EPR effect is not the main driving force in the accumulation of nanoparticles in human tumors [7,8,9]. The EPR effect is much more pronounced in rodents with rapidly developing tumors that do not have time to form a normal vascular network compared to large solid tumors in humans [10]. In this regard, there is an urgent need to develop drugs that differ in the mechanism of action from traditional drugs based on nanoparticles, such as the liposomal form of doxorubicin (Myocet and Caelyx) [11,12]. Specifically, targeted drug delivery systems (DDS) have improved characteristics in terms of accumulation in the tumor, but their use to this day remains controversial in biomedicine. In particular, to date, there are no **targeted** nanomedications approved by the FDA for the treatment of diseases [13].

Targeted drug delivery is an important tool in the treatment of HER2-positive tumors, which are known for their aggressive nature. The human epidermal growth factor receptor 2 (HER2 receptor) is an important target for drug delivery in the treatment of HER2-positive tumors [14]. This receptor is found on the surface of certain types of cancer cells, and when activated by drugs, it can help to stop the growth and spread of these tumors. Drugs that target the HER2 receptor can be used alone or in combination with other treatments to improve patient outcomes. Studies have shown that using drugs that specifically target the HER2 receptor can help to reduce tumor size and slow or stop tumor growth [15,16]. Additionally, targeting this receptor has been shown to increase survival rates and reduce recurrence rates in patients with HER2-positive tumors [17].

In order to enhance the efficacy of targeted nanoagents as a new generation of diagnostic and therapeutic tools, various approaches are being developed, which are aimed at changing both the architecture of nanoagents [18,19] and the methods of their systemic administration in order to achieve maximum efficiency [20,21,22]. Here, we describe a combination of these approaches, namely, the development of both a new design of a polymer nanocarrier and a method for its systemic delivery to the tumor site.

In particular, we developed polymer nanocapsules for oncotheranostics and a method for their targeted two-step delivery to HER2-overexpressing tumors in immunocompetent BALB/c mice.

These capsules loaded with the diagnostic dye, Nile Blue, and the chemotherapeutic compound, doxorubicin, were used for the two-step delivery to HER2+ tumors using the concept of tumor pre-targeting with targeting molecules using barnase/barstar “bacterial superglue” [23,24,25,26]. First, the tumor was pre-targeted with an anti-HER2 targeting molecule fused with barstar, and then barnase-modified nanocapsules were injected into the organism. We have shown that such a two-step delivery is much more effective than one-step delivery in terms of both imaging and tumor therapy. In particular, using the unique HER2-positive tumor of BALB/c mice developed by us in this work, we demonstrate that the developed two-step DDS based on barnase/barstar showed superior efficiency in image-guided cancer therapy: HER2-positive tumors’ treatment tests showed the tumor growth inhibition index TGI = 68.4% for one-step DDS and TGI = 94.9% for two-step DDS.

## 2. Materials and Methods

### 2.1. PLGA Nanoparticle Synthesis and Characterization

PLGA nanoparticles were synthesized, characterized, and conjugated to proteins, as described by us previously [27].

### 2.2. Cell Culture

EMT6/P and EMT-HER2 cells (Shemyakin-Ovchinnikov Institute RAS, Molecular Immunology Laboratory collection) were cultured in DMEM medium (HyClone, Logan, UT, USA) supplemented with 10% FBS (HyClone, Logan, UT, USA), penicillin/streptomycin (PanEko, Moscow, Russia), and 2 mM L-glutamine (PanEko, Moscow, Russia) under a humidified atmosphere at 37 °C and 5% CO_2_.

### 2.3. EMT-HER2 Cells

The DNA fragments coding for the full-size human HER2 receptor (P04626) were synthesized (GeneCust, Boynes, France) and cloned into the pLV2 lentiviral vector (Clontech, San Jose, CA, USA) under the control of the EF1a promoter. HEK293T cells were co-transfected with one of the viral plasmids and the set of packaging third generation plasmids. Supernatants containing viruses were collected at 48 h post-transfection. The titer of lentivirus preparations was determined using Lenti-X p24 ELISAs (Clontech, San Jose, CA, USA). EMT6/P cells were transduced with as-obtained lentiviruses and 48 h after, cells were seeded onto a 96-well plate at 1 cell per well to obtain single colonies. Single colonies were further grown in 25 cm^2^ flasks and then analyzed with flow cytometry for the expression of the HER2 receptor. The cell line with moderate receptor expression was selected for further in vivo experiments.

### 2.4. Flow Cytometry

Cells were washed with PBS and resuspended in 300 μL of PBS with 1% BSA at a concentration of 10^6^ cells/mL. Cells were labeled with Trastuzumab-FITC in a final concentration of 2 μg/mL, washed, and analyzed in the FL1 channel (excitation laser—488 nm; emission filter—530/30 nm) using a NovoCyte 3000 VYB flow cytometer (ACEA Biosciences, San Diego, CA, USA) in BL1 channel (excitation laser 488 nm, emission filter 530/30 nm).

### 2.5. Tumor-Bearing Mice

Female BALB/c mice of 22–25 g weight were purchased from the Puschino Animal Facility (Shemyakin-Ovchinnikov Institute of Bioorganic Chemistry Russian Academy of Sciences, Pushchino branch of the Institute, Pushchino, Russia) and maintained at the Vivarium of the Shemyakin-Ovchinnikov Institute of Bioorganic Chemistry Russian Academy of Sciences (Moscow, Russia). All procedures were approved by the Institutional Animal Care and Use Committee (IACUC) of the Shemyakin-Ovchinnikov Institute of Bioorganic Chemistry Russian Academy of Sciences (Moscow, Russia) according to the IACUC protocol #299 (1 January 2020–31 December 2022).

The animals were anesthetized with”a mixture of tiletamine/zolazepam/xylazine at a dose of 20/20/2 mg/kg (Zoletil (Virbac, Carros, France) and Rometar (Bioveta, Ivanovice na Hané, Czech Republic)).

Female BALB/c mice (18–22 g) were injected with 4 × 10^6^ EMT-HER2 cells in 100 µL of full culture medium in the right flank to create tumors. The tumor size was measured with a caliper using the formula V = width^2^ × length/2.

### 2.6. Cryosections

Mice were sacrificed with cervical dislocation, and cryosections of tumor tissue were obtained using FSE cryostat (Thermo Scientific, Kalamazoo, MI, USA). Cryosections were fixed with 4% paraformaldehyde, permeabilized with PBS supplemented with 0.01% TritonX-100, and labeled with Hoechst33342 and anti-HER2 IgG-FITC in PBS supplemented with 1% BSA for 30 min, followed by PBS washing, and mounted using VECTASHIELD Antifade Mounting Medium (Vector Laboratories, Newark, NJ, USA).

### 2.7. Confocal Laser Scanning Microscopy

EMT6/P and EMT-HER2 cells were labeled Trastuzumab-FITC at 2 µg/mL in PBS with 1% BSA for 30 min at +4 °C. The confocal microscopy images of cells were obtained with an FV3000 laser-scanning confocal microscope (Olympus Optical Co Ltd., Tokyo, Japan) using LUCPLFLN 20× objective (20× magnification, 0.45 numerical aperture) with a 488 nm laser and a GaAsP detector (542 V) (500–600 nm).

The confocal microscopy images of cryosections were obtained with an FV3000 laser-scanning confocal microscope (Olympus Optical Co. Ltd., Tokyo, Japan) using a UPLSAPO 40 × 2 objective (40× magnification, 0.95 numerical aperture) with a 405 nm laser and a GaAsP detector (447 V) (430–470 nm) for Hoechst33342 detection and a 488 nm laser and a GaAsP detector (545 V) (500–600 nm) for FITC detection.

### 2.8. In Vivo Imaging

In vivo imaging was performed with a LumoTrace FLUO bioimaging system (Abisense LLC, Sochi, Russia) as follows: mice were anesthetized 4 h after the injection of nanoparticles and imaged with fluorescence excitation at a λ_ex_ = 730 nm and 780 nm long-pass filter.

### 2.9. Red Blood Cell Hemolysis

Red blood cells (RBC) were isolated from fresh mouse whole blood, cleansed of serum with centrifugation, and resuspended in phosphate-buffered saline (PBS) to obtain a final 5% hematocrit level. The proteins under investigation were incubated with RBC for 2 h at room temperature. Next, red blood cells were centrifuged for 3 min at 500× *g*, and the absorbance of the supernatant was measured. As the positive control (equal to complete lysis), the sample of RBC lyzed with H_2_O was used. As the negative control (= spontaneous lysis), the sample of RBC incubated with PBS was used. The quantity of the released hemoglobin in the samples was measured at 540 nm and further presented in percent from the optical density of samples lyzed with H_2_O (equal to full lysis).

### 2.10. Agglutination Study

The hemagglutination study was performed on 96-well U-shaped plates. The fresh whole blood was twice washed with PBS with 1% BSA and resuspended to obtain a final 1% concentration of RBC. The proteins under investigation were then added to the RBC samples, and 40 µL of the sample was then placed into the well of the 96-well plate. The samples were incubated for 1 h at room temperature. The RBC sample without proteins was used as negative control (in the absence of agglutination). For the positive control, the sample of RBC was incubated with rat anti-mouse RBC IgG (TER-119, 50 µg/mL) for 10 min and washed of non-bound antibodies. Next, a secondary goat anti-rat antibody was added to obtain a final 10 µg/mL IgG concentration. This sample, with antibody-mediated agglutination, was used as the positive control. When agglutination occurs, the red blood cells do not form a spot on the well bottom, forming a film over the surface of the well.

### 2.11. Immunogenicity Study

To purify proteins produced in E. coli from lipopolysaccharides, High Capacity Endotoxin Removal Spin Columns, 0.5 mL (Pierce), were used according to the protocol recommended by the manufacturer. For all manipulations, we used laboratory plastic marked “sterile” and “non-pyrogenic”. At 10 min before protein injection, animals were anesthetized with a mixture of Zoletil and Rometar at a dose of 25/5 mg/kg. On days 1, 3, 5, 7, 9, 11, and 13, mice were injected intraperitoneally with 10 μg of proteins in 100 μL of sterile pyrogen-free PBS. Prior to and 21 days after the first injection, a blood sample was taken, and the serum was isolated by centrifugation for 10 min at 400× *g* at 15 °C after a 30 min incubation at room temperature for blood clotting. The number of protein-specific antibodies was analyzed by enzyme-linked immunosorbent assay.

### 2.12. ELISA

The protein under investigation (barnase or barstar) in an amount of 0.5 μg per well was sorbed overnight at +4 °C in 100 μL of carbonate buffer (4 mM Na_2_CO_3_, 50 mM NaHCO_3_, pH 9.2) in the wells of 96-well ELISA plates. Then, the wells were washed twice with 200 μL of PBS with 0.05% Tween-20. Diluted mouse blood serum in 100 μL PBS with 1% BSA was then added to the wells and incubated for 1 h at room temperature. Then, the wells were washed twice with 200 μL of PBS with 0.05% Tween-20. The wells were added with 100 μL of anti-mouse antibodies with alkaline phosphatase in 100 μL of PBS with 1% BSA and incubated for 1 h at room temperature. Then, the wells were washed three times with 200 μL of PBS with 0.05% Tween-20. Thereafter, p-nitrophenyl phosphate was added to the wells at a concentration of 10 g/L in glycine buffer (0.1 M glycine, 1 mM MgCl_2_, pH 2). The reaction was halted with a 0.1 M NaCl solution, pH 10.4, and the absorption of the samples was recorded at a wavelength of 405 nm.

## 3. Results

### 3.1. Design of the Experiment: One-Step DDS vs. Two-Step DDS In Vivo

We synthesized polymer nanocapsules based on a copolymer of lactic and glycolic acids, PLGA. These nanocapsules were loaded with imaging dye (Nile Blue) and a chemotherapeutic compound (doxorubicin), as described in detail by us previously [27]. These PLGA nanocapsules were delivered to HER2-overexpressing cancer cells in vivo using the HER2-recognizing scaffold polypeptide DARPin9_29 through two different drug delivery strategies (DDS), namely one-step DDS and two-step DDS (Figure 1). DARPin9_29 is a small scaffold protein (14 kDa) that recognizes the extracellular domain of HER2 receptor with high selectivity (K_aff_ = 3.8 nM).

The synthesis of PLGA nanoparticles is based on a double “water-oil-water” microemulsion technique (Figure 2a). Doxorubicin is incorporated into the nanoparticle structure through the first “water” phase, while Nile Blue is incorporated into the “oil” phase [27].

For one-step DDS, polymer PLGA nanoparticles were conjugated to HER2-recognizing scaffold polypeptide DARPin9_29 using carbodiimide chemistry toobtainPLGA-DARPin9_29. For two-step DDS, the pre-targeting of HER2-overexpressing cells was realized using DARPin9_29 fused with barstar (Bs-DARPin9_29) with the subsequent injection of nanoparticles conjugated to barnase (PLGA-Bn). Barnase and barstar are small bacterial proteins (12 and 10 kDa) acting as “bacterial superglue” due to their extremely high affinity constant (K_d_ = 10^−14^ M).

We previously performed a direct comparison of the efficiency of these two strategies for drug delivery to HER2-overexpressing cells in vitro and demonstrated that two-step DDS significantly outperforms one-step delivery in terms of HER2-positive cell targeting and doxorubicin-induced cytotoxicity.

Based on our previously obtained results, we theorized that two-step DDS was more efficient than one-step DDS in vivo for HER2-positive tumor imaging and treatment. To evaluate the efficiency of barnase/barstar-mediated PLGA two-step delivery to HER2-positive tumors, we developed the HER2-overexpressing tumor model for immunocompetent mice, namely, the most popular BALB/c mice. The mouse mammary cancer cells EMT6/P were transduced with the HER2 gene and a single clone (EMT-HER2) was selected that was able to form HER2-overexpressing tumors in BALB/c mice with 100% efficiency and reproducibility. Using this allograft tumor model, we performed the comparison of one-step and two-step DDS based on the barnase/barstar interface and PLGA nanocarrier. Living imaging tests and a tumor growth dynamic study confirmed that the dose of PLGA-DARPin9_29 that was virtually non-effective for HER2+ tumor imaging and therapy in a one-step strategy led to effective diagnostics and treatment with a two-step strategy.

### 3.2. Synthesis and Characterization of Polymer Nanocarriers for Targeted Delivery to HER2-Overexpressing Cancer Cells

Polymer oncotheranostic nanoparticles were synthesized as previously described by us in detail [27]. Namely, PLGA nanoparticles possessing diagnostic and therapeutic properties were synthesized using the water–oil–water microemulsion technique, as shown in Figure 2a and described in [27]. The hydrophilic chemotherapeutic drug, doxorubicin, was loaded into the first water phase and the diagnostic hydrophobic dye, Nile Blue, was loaded into the oil phase. We previously demonstrated that the incorporation of doxorubicin incorporation was found to be 0.9 nmol doxorubicin per 1 mg of particles. Polyvinyl alcohol was mixed with chitosan oligosaccharide lactate to stabilize zwitterionic nanoparticles in saline solutions [27].

Synthesized PLGA nanoparticles possessing diagnostic and therapeutic functions were characterized by scanning electron microscopy, which proved that nanoparticles possess a spherical form with narrow size distribution (Figure 2b). The physical size distribution of synthesized nanoparticles obtained by scanning electron microscopy image processing was found to be equal to 206 ± 52 nm (Figure 2c).

Next, the synthesized PLGA nanoparticles with Nile Blue and doxorubicin were conjugated to:

(i) anti-HER2 scaffold protein DARPin9_29 to obtain PLGA*DARPin9_29;

(ii) barnase to obtain PLGA-Bn for self-assembly with the fusion protein of DARPin9_29 and barstar, Bs-DARPin9_29, to obtain self-assembled structures PLGA-Bn/Bs-DARPin9_29 on the surface of HER2-overexpressing cells, as described in detail in [27].

### 3.3. Development of HER2-Overexpressing Allografts in Immunocompetent BALB/c Mice

To assess the efficacy of one-step and two-step DDS for oncotheranostic PLGA delivery in vivo, we developed a mouse tumor model with the human oncomarker HER2 overexpression. For this, mouse mammary cancer cells EMT6/P were transduced with transmembrane receptor HER2 via a lentiviral transfection system, and a single clone of the resultant cells was selected and grown. The HER2 expression in the obtained cells, EMT-HER2, was confirmed with flow cytometry, as shown in Figure 3a and confocal laser scanning microscopy, as shown in Figure 3b. EMT-HER2 cells were stained with the anti-HER2 monoclonal antibody, trastuzumab, and conjugated to fluorescein isothiocyanate, FITC (anti-HER2 IgG-FITC). Data presented in Figure 3 confirm that EMT-HER2 is specifically and selectively labeled with anti-HER2 IgG with no non-specific binding.

Next, these EMT-HER2 cells were s.c. injected into the right flank of BALB/c mice in full culture medium and the dynamic of tumor growth was monitored with caliper measurements. After the tumor size reached ~200 mm^3^, the cryosections of the tumor were performed and stained with anti-HER2 IgG and Hoeschst33342, as shown in Figure 4. Data presented in Figure 4 confirm that these cells stably express receptor HER2 both in vitro and in vivo in immunocompetent BALB/c mice.

### 3.4. In Vivo Bioimaging: HER2-Positive Allograft Visualization with One-Step and Two-Step DDS

To compare the effectiveness of targeted two-step and one-step therapy mediated by scaffold polypeptides DARPin9_29 and proteinaceous barnase/barstar interface, the following in vivo tests were performed. Obtained EMT-HER2 tumors were used for the bioimaging tests and tumor growth inhibition studies. Mice were divided into three groups that received the following injections:

(i) For the study of one-step DDS efficacy in vivo, the injection of the conjugate of PLGA nanoparticles with DARPin9_29 was performed, namely, mice received an i.v. injection of 500 µg of the conjugate of PLGA-DARPin9_29 on days 8 and 10 of the treatment;

(ii) For the study of two-step DDS efficacy in vivo, the injection of DARPin9_29-Bs followed by the injection of the conjugate of PLGA with barnase, PLGA-Bn was used. Namely, mice received an i.v. injection of 150 µg of DARPin9_29-Bs and, 2 h later, an i.v. injection of 500 µg of the conjugate of PLGA-Bn on days 8 and 10 of the treatment;

(iii) the control group received no treatment.

The accumulation of PLGA nanoparticles in the tumor area was monitored 4 h after the nanoparticles injection with the Lumotrace FLUO bioimaging system, as shown in Figure 5. We demonstrated that the two-step DDS was much more effective in terms of nanoparticle accumulation in tumors for bioimaging purposes.

### 3.5. In Vivo Therapy: One-Step DDS vs. Two-Step DDS for HER-Positive Tumor Growth Inhibition

To evaluate the efficacy of two-step DDS in comparison with one-step delivery for HER2-overexpressing tumor treatment, we evaluated tumor growth dynamics in all experimental groups (n = 3 for each group). According to the treatment scheme, mice received a total of 0.9 nmol of doxorubicin (by two sequential injections of 500 µg of nanoparticles equal to a total of 1 mg of nanoparticles). The results are presented in Figure 6.

The tumor growth inhibition indexes at 21 day calculated as %TGI = (1 − {T_t_/T_0_}/{C_t_/C_0_})/(1 − {C_0_/C_t_}) × 100, where T_t_ = the median tumor volume of treated at time t, T_0_ = the median tumor volume of treated at time 0, C_t_ = the median tumor volume of control at time t, and C_0_ = the median tumor volume of control at time 0 that were found to be equal to TGI_1_ = 68.4% for one-step DDS and TGI_2_ = 94.9% for two-step DDS. The obtained data show that the two-step delivery of the same dose of doxorubicin-loaded nanoparticles was much more effective than one-step delivery, both in terms of diagnostics and therapy.

### 3.6. Biosafety Aspects: Barnase and Barstar Hemotoxicity Study

During the development of therapeutic drugs, in particular those of a protein origin, one of the key biocompatibility parameters to be considered is the hemotoxicity and immunogenicity of a given drug. These studies should be carried out to assess the possibility of multiple administrations of the drug and/or repetition of the course of therapy. In order to assess the possible risks associated with the systemic repeated administration of structures containing the proteins barnase and barstar, the hemotoxicity and immunogenicity of these proteins were investigated separately.

To study hemotoxicity, in particular, possible hemolysis and hemagglutination, mouse erythrocytes were incubated with proteins, namely, barnase or barstar at various concentrations. The amount of released hemoglobin was assessed by measuring the absorbance at a wavelength of 540 nm, and the value obtained was expressed as a percentage of the absorbance of the positive control sample (complete lysis), as described in detail in the Section 2. It was shown that all tested proteins in a wide range of concentrations do not cause the hemolysis of erythrocytes (Figure 7). Hemagglutination studies were performed in U-shaped 96-well plates. For this, mouse red blood cells were incubated with proteins: barnase or barstar at various concentrations in U-shaped plates. In the case of hemagglutination, blood cells did not settle at a point in the center of the well and formed a film over the entire surface. It was shown that all tested proteins in a wide range of concentrations do not cause hemagglutination (Figure 7).

### 3.7. Biosafety Aspects: Immunogenicity Study of Barnase and Barstar

To study the immunogenicity of barnase and barstar, BALB/c mice (18–22 g) were injected intraperitoneally with 10 μg of proteins in 100 μL of sterile pyrogen-free PBS on study days 1, 3, 5, 7, 9, 11, and 13. Proteins were pre-purified from lipopolysaccharides using Pierce High Capacity Endotoxin Removal Spin Columns, 0.5 mL. A group of animals that were injected with PBS without proteins was used as a “negative” control. As a “positive” control, a group of animals that were injected with proteins according to the same scheme was used, but at the same time, on day 1, a protein was injected in a mixture with 50 μL of complete Freund’s adjuvant, and on day 13, a protein mixed with 50 μL of incomplete Freund’s adjuvant was injected. Before and 21 days after the first injection, the blood samples were taken, serum was isolated, and the number of protein-specific antibodies in the serum was analyzed in the enzyme-linked immunosorbent assay format, as described in the Section 2.

The results of the enzyme-linked immunosorbent assay, namely, the number of antibodies, depending on the serum dilution for mice injected with the proteins barnase or barstar for the main and control groups, are shown in Figure 8b. The data presented in Figure 8b indicate the absence of a specific immune response in mice for the tested proteins on day 21 after the first injection even with the “boosting” the immune response by injecting proteins in a mixture with complete and then incomplete Freund’s adjuvant. At the same time, the weight of the animals (Figure 8a) did not change significantly throughout the entire experiment for all experimental groups. The obtained data indicate the possibility of multiple administrations of the studied proteins without serious risks associated with the specific B-cell immune response of the organism.

## 4. Discussion

The development of targeted drug delivery systems (DDS) for chemotherapeutic drugs for clinical use necessitates a comprehensive evaluation of various parameters to ensure their maximum efficiency. Unfortunately, there are still no **targeted** nanoparticles that have been approved for clinical use in patients. To improve the efficacy of targeted DDS and reduce their systemic toxicity, there is an urgent need for the optimization of nanoagents and extensive fundamental research to develop the most specific nanostructures with minimal toxicity. In addition, there is a need to develop novel strategies for their systemic administration in vivo. These challenges demand a multidisciplinary approach and novel strategies to address the challenges associated with targeted drug delivery, such as the need to develop innovative nanostructures with improved biocompatibility, targeted uptake, and the controlled release of therapeutic agents. The development of targeted DDS for chemotherapeutic drugs must involve an extensive investigation of various parameters, such as the design and optimization of nanoparticles, the development of methods for their systemic administration in vivo, and the assessment of their efficacy in reducing systemic toxicity.

The development of new nanoparticles for targeted drug delivery can be improved through the concept of pre-targeting the tumor site. By introducing a large amount of a targeted non-toxic compound prior to injecting the second nanoparticle-based cytotoxic module, it is possible to increase the affinity of the nanoparticles and thus achieve more effective drug delivery. This strategy makes it possible to achieve a significantly higher efficiency of nanoparticles for both diagnostics and therapy with the same administered doses of nanoparticles.

The concept was confirmed in this work using the barnase*barstar protein pair as molecular mediators between the first and second modules. Namely, barstar-DARPin9_29 was used as the first component of the two-step DDS and nanoparticles modified with barnase, which served as the second component.

As shown in Table 1, the barnase*barstar protein pair outperforms other two-step systems, which makes it a unique tool for the design of multifunctional biomedical products. Barstar (10 kDa) is a natural inhibitor of bacterial ribonuclease barnase (12 kDa) [28]. These proteins have an extremely high binding affinity (association constant K_aff_~10^14^ M^−1^) and fast interaction kinetics (rate constant of complex formation k_on_~10^8^ M^−1^s^−1^). At the same time, these proteins are not present in mammals, thus making them excellent candidates for use in the bloodstream without any interaction with endogenic components of blood [29,30]. Moreover, here we show that the barnase*barstar protein pair is not immunogenic in mice.

Along with the method of systemic administration, the composition of the nanocarrier is no less important for the effectiveness of the DDS. A co-polymer of fully biocompatible and biodegradable lactic and glycolic acids, PLGA, was used as the platform for the design of this two-step DDS. PLGA is the most popular polymer commonly used for biomedical and fundamental research applications [15,31]. PLGA has already been approved by the FDA for some therapeutic purposes [32,33] and has demonstrated remarkable results in clinical trials as an excellent candidate for drug delivery and treatment [34,35].

These PLGA nanoparticles were loaded with doxorubicin for treatment options, and the fluorescent dye, namely, Nile Blue, was incorporated into the PLGA structure for diagnostic applications. Nile Blue is a fluorescent dye from the benzophenoxazine family with high fluorescence, high quantum yield, and excellent photostability [36]. The maximum excitation of the dye in dimethyl sulfoxide is 636 nm, and the maximum emission is 669 nm, thus entering the transparency window of biological tissues and making this dye promising for imaging applications in vivo [36]. Due to its lipophilic structure, Nile Blue is already used for several biological applications, e.g., for histology in vitro. Moreover, several in vivo studies have demonstrated the ability of this dye to accumulate in tumor cells after i.v. administration [37,38]. Despite the above-mentioned advantages and low cost, Nile Blue has been unjustly underestimated in biology with only a limited number of studies related to its usage. Here, we have shown that Nile Blue is a very effective diagnostic tool for targeted DDS, and we believe that its further application will be extended to the development of other drug delivery systems, not only to the HER2 oncomarker.

The concept of targeted drug delivery suggests several advantages over standard chemotherapy, such as the decrease in the required dose of a drug, the improved drug penetration into the tumor, and the reduction of side effects [39]. However, the use of traditional targeting molecules for the therapeutics delivery, namely, the use of monoclonal antibodies, often leads to a wide spectrum of undesirable effects: (i) the considerable size of antibodies (150 kDa, 7–14 nm) often does not allow the modification of the nanoparticle surface with the required number of IgG molecules for the efficient delivery to cells; (ii) the post-translational modifications of IgGs require biotechnological production in mammals, which is time-consuming and expensive; (iii) the constant domains of the heavy chains have effector functions that may lead to phagocytosis without participating in the selective target recognizing or can cause unwanted immunomodulation in vivo; and (iv) the presence of cysteines in the antibody molecule and glycosylation, which play an important structural role [40,41,42,43,44].

The artificial scaffold polypeptides were used in this study to target the PLGA nanoparticles toward cancer cells instead of the traditionally used full-size IgGs. The targeted scaffolds of non-immunoglobulin nature obtained by phage, cellular, or ribosomal display technologies seem to be more effective tools for the delivery of nanoparticles and other substances to the cancer cells in the tumor site. The most popular synthetic scaffold proteins for targeted delivery are DARPins (synthetic derivatives of cytoskeleton protein of drosophila—ankyrin) [45,46,47,48,49], monobodies (derivatives of human fibronectin FN3 domain) [50], anticalins (derivatives of lipocalins) [51], avimers (derivatives of extracellular receptor A-domain) [42], affibodies (derivatives of highly stable domain B of staphylococcal protein A) [31], and others. Designed ankyrin repeat protein, or DARPin, was used in this study for the following reasons: it has a small size (14 kDa); high affinity to the molecular target, namely, receptor HER2 (K_D_ = 3.8 nM); low immunogenicity; exceptional thermodynamic stability; and an absence of cysteines in its structure [45,52,53,54,55]. Equally important is the ease of large-scale biotechnological production, in contrast to that of full-size antibodies [45,56]. All these properties simplify genetic engineering and the creation of multispecific fusion proteins, which allow not only targeting different structures to the cells with specific molecular profiles but also realize their own diagnostic and therapeutic functions [15,46,49,57,58]. Here, we used DARPin9_29, which specifically recognizes HER2 receptors on the cancer cell surface. DARPin9_29 was genetically fused with barstar, which allowed us to realize the two-step delivery system through the barnase*barstar interface.

Thus, using barstar-DARPin9_29 and barnase-conjugated polymer PLGA nanoparticles loaded with the fluorescent dye, Nile Blue, and chemotherapeutic drug, doxorubicin, we showed successful bioimaging and tumor elimination in vivo. Namely, the developed two-step DDS based on barnase*barstar showed superior efficiency in image-guided cancer therapy: HER2-positive tumors’ treatment tests showed the tumor growth inhibition index TGI = 68.4% for one-step DDS and TGI = 94.9% for two-step DDS. The obtained results demonstrate the significant superiority of two-step drug delivery over classical one-step delivery. We expect these data to change the paradigm in next-generation drug development and enable researchers to focus on multi-step targeted drug delivery to reduce the required dose in cancer therapy.

**Table 1 pharmaceutics-15-00833-t001:** The interfaces used for the drug delivery systems with pre-targeting steps (✓—advantages, ✗—drawbacks).

Delivery System	Immunogenicity	Steric Hindrance	K_a_	Representation in Mammals
Barnase*barstar	✓ Both proteins are not immunogenic (this study).	✓ Proteins are comparable in size (12 and 10 kDa), and therefore steric hindrances should not arise [28].	10^14^ M^−1^ [30]	✓ Both proteins were isolated from bacteria and are not represented in mammals [30].
Streptavidin*biotin	✗ Streptavidin is highly immunogenic [59].	✗ The significant difference in the size (56 kDa and 244 Da) of the molecules can cause steric hindrance: if biotin is bound to a non-smooth surface, then streptavidin will not be able to recognize it. This imposes restrictions on the use of this system in nanomedicine [60,61,62].	10^15^ M^−1^ [63]	✗ Biotin, or vitamin H, is presented in the blood of mammals, which may cause obstacles to the appropriate interaction of streptavidin*biotin [64,65,66].
Hapten* antibody	✗ Antibodies are immunogenic and may have effector functions, which make them not the best candidate for long-term treatment [67].	✗ IgG is 150 kDa protein, while hapten usually is a low-molecular compound, so steric difficulties may arise when the components interact [67,68].	10^5^–10^10^ M^−1^ [69]	✗ Antibodies are presented in blood and have effector functions as critical participants in the immune defense [41].
Nucleic acids: (i) DNA*DNA (ii) RNA*RNA (iii) mirror-imaged oligonucleotides (iv) phosphorodiamidate morpholino oligomers (v) peptide nucleic acids (vi) locked nucleic acid	✓ Nucleotides as natural molecules are not immunogenic [70]. ✓ Some mirror-imaged oligonucleotides, phosphorodiamidate morpholino oligomers, and peptide nucleic acids are not immunogenic [71,72,73,74].	✗ Nucleotides are small molecules (less than 500 Da). Since the entire sequence of nucleotides is essential for recognition purposes, steric hindrance is a common problem in recognition processes.	Depending on the base pair number	✗ The presence of nucleases in the serum is an obstacle in the development of the system based on oligonucleotides due to the fast degradation [70]. ✓ Due to their artificial origin, mirror-imaged oligonucleotides, phosphorodiamidate morpholino oligomers and peptide nucleic acids are resistant to degradation by nucleases, and peptide and locked nucleic acids are also resistant to protease digestion [72,74,75,76,77].
Click chemistry	✓ Molecules used in bio-orthogonal chemistry are supposed to be not highly immunogenic [78].	✗ Molecules used in bio-orthogonal chemistry are small, which can cause steric hindrance.	✓ ✗ Covalent bonding is stronger than affinity interaction but is not reversible.	✓ Molecules used in bio-orthogonal chemistry are not represented in mammals [75].
SpyTag/SpyCatcher	N/A	✓ Steric hindrances should not arise [79].	✓ ✗ Covalent bonding is stronger than affinity interaction but is not reversible.	✓ Both proteins were isolated from bacteria and modified by bioengineering and are not represented in mammals [80].

## 5. Conclusions

Here, we present a two-step drug delivery system (DDS) for theranostic applications based on the barnase*barstar proteinaceous interface and PLGA nanocarriers. This “molecular glue” offers a versatile “lego” approach for nanoparticle biomodification, allowing for a two-step targeted delivery in vivo. This pre-targeting concept can reduce doses of drugs to obtain the same therapeutic effect, reducing systemic toxicity and side effects. This system outperforms existing technologies and may promote the development of new-generation drug delivery systems for cancer diagnosis and treatment.

## Figures and Tables

**Figure 1 pharmaceutics-15-00833-f001:**
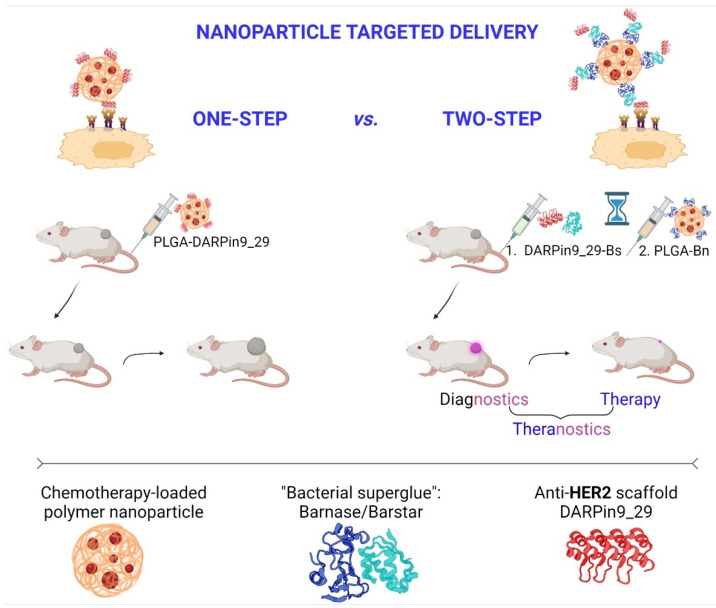
Two-step targeted delivery of polymer oncotheranostic nanoparticles is more effective than one-step delivery for image-guided HER2-overexpressing breast cancer treatment: scheme of the experiment.

**Figure 2 pharmaceutics-15-00833-f002:**
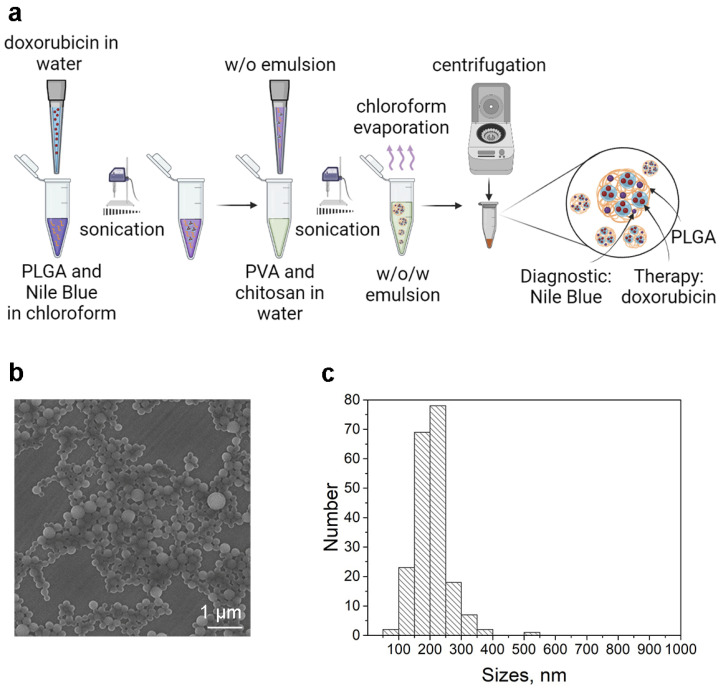
Synthesis and characterization of PLGA nanoparticles for targeted drug delivery. (**a**) Scheme of the synthesis. (**b**) Scanning electron microscopy of PLGA nanoparticles. (**c**) Physical size distribution obtained through SEM image processing.

**Figure 3 pharmaceutics-15-00833-f003:**
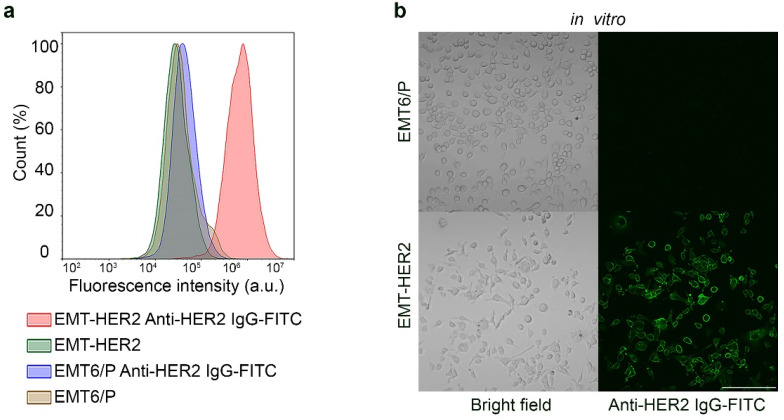
In vitro cell analysis of EMT-HER2 cells. (**a**) Flow cytometry analysis on cell labeling with anti-HER2 IgG-FITC. Initial EMT6/P and cells with HER2 (EMT-HER2) were labeled with anti-HER2 IgG-FITC and analyzed in the fluorescence channel corresponding to FITC fluorescence. (**b**) Confocal laser scanning microscopy of EMT6/P and EMT-HER2 cells labeled with Trastuzumab-FITC. Left panels—bright-field images; right panels—confocal images of the cells. Scale bar, 250 µm.

**Figure 4 pharmaceutics-15-00833-f004:**
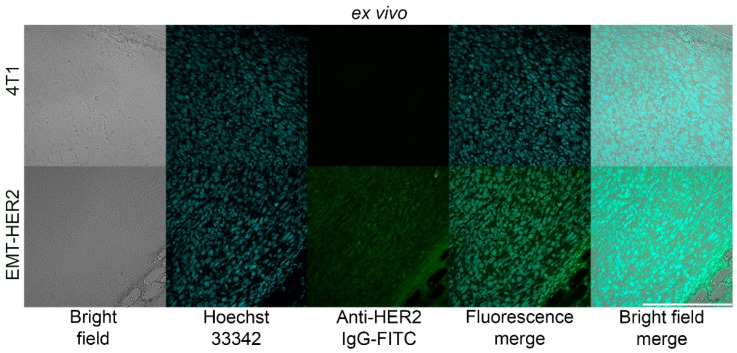
Ex vivo imaging of HER2 expression in EMT-HER2 tumors. Tumors were excised, and cryosections were stained with Hoechst33342 and anti-HER2-FITC IgG followed by confocal laser scanning microscopy analysis in bright field and fluorescence channels, corresponding to Hoeschst33342 and FITC fluorescence. 4T1 tumors with no HER2 expression served as the negative control. Scale bar, 250 µm.

**Figure 5 pharmaceutics-15-00833-f005:**
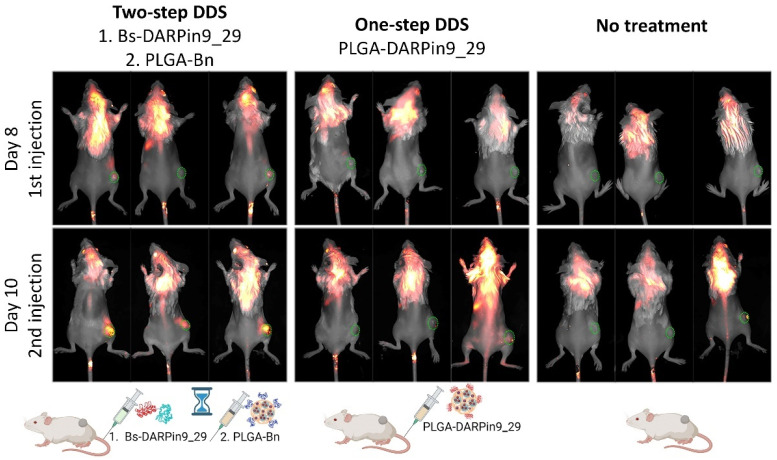
Imaging of BALB/c mice with EMT-HER2 tumors after the first and second injections of the one-step DDS, PLGA*DARPin9_29 particles, or two-step DDS, Bs-DARPin9_29 and PLGA-Bn, in comparison with the control group. Green circles indicate tumor area. Imaging was performed with LumoTrace FLUO bioimaging system with fluorescence excitation using a 730 nm and 780 nm long-pass filter, thus registering the PLGA particles’ fluorescence.

**Figure 6 pharmaceutics-15-00833-f006:**
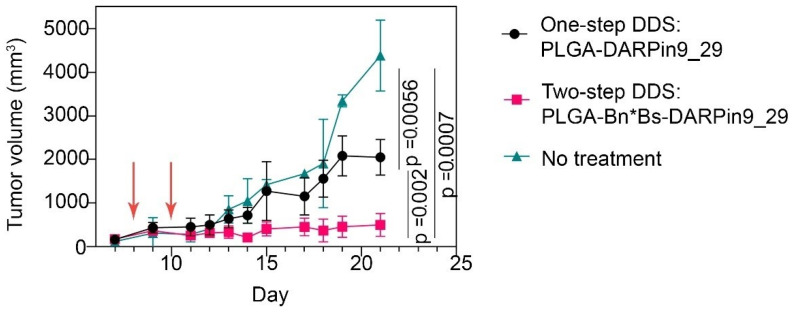
EMT-HER2 tumor growth dynamics under the treatment with one-step DDS, PLGA*DARPin9_29 particles, or two-step DDS, Bs-DARPin9_29 and PLGA-Bn (n = three mice for each group). Red arrows indicate the nanoagent injections.

**Figure 7 pharmaceutics-15-00833-f007:**
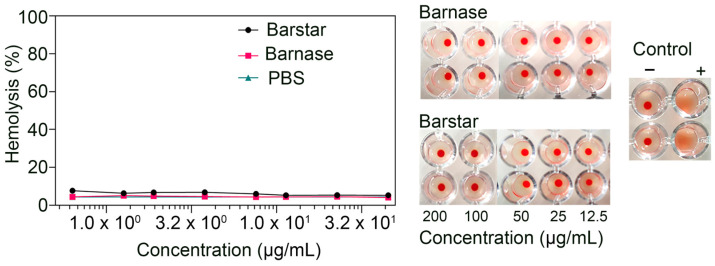
Hemotoxicity study of barnase and barstar. The intensity of erythrocyte lysis after the incubation with barnase or barstar at various concentrations is presented as a percentage of the positive control. A sample of erythrocytes lysed with distilled water was used as a positive control, and a sample of erythrocytes incubated in PBS was used as a negative control. The data are accompanied by the agglutination study. RBCs were sequentially incubated with rat anti-RBC antibodies (TER119) and goat anti-rat antibodies were used as a positive control. A sample of RBCs incubated in PBS was used as a negative control.

**Figure 8 pharmaceutics-15-00833-f008:**
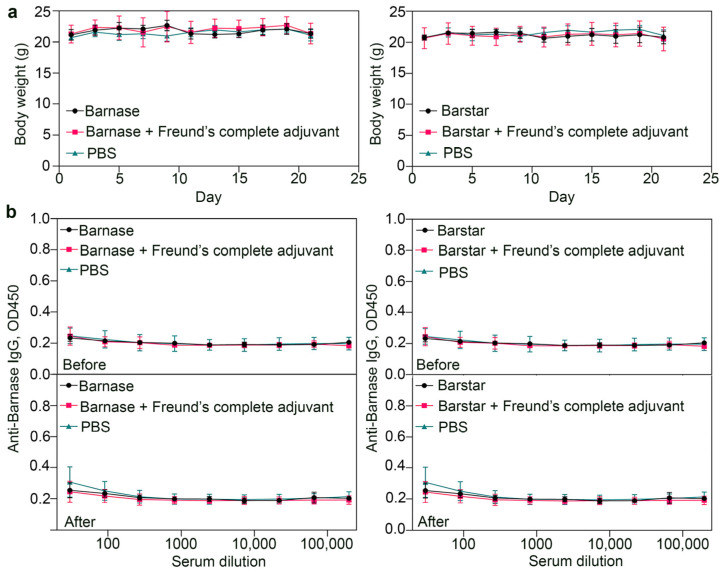
Immunogenicity study of barnase and barstar. (**a**) Body weight dynamics after the sequential injections of barnase or barstar for the experimental and control animal groups (n = 3–5 mice). (**b**) Study of a protein-specific immune response for the proteins barnase and barstar. BALB/c immunocompetent mice were used as experimental animals. The graphs show the absorption of the phosphatase substrate, corresponding to the quantity of antibodies specific to barnase or barstar, depending on the blood serum dilution in blood samples before and after the sequential administration of the corresponding proteins.

## Data Availability

All data are presented within the manuscript.

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
