# Peer review of "Targeted Two-Step Delivery of Oncotheranostic Nano-PLGA for HER2-Positive Tumor Imaging and Therapy In Vivo: Improved Effectiveness Compared to One-Step Strategy"

_pharmaceutics, 2023, doi:10.3390/pharmaceutics15030833_

Round 1

Reviewer 1 Report

In this manuscript, the authors described a novel polymeric nanoparticle to deliver doxorubicin into HER-2-positive tumors with a theranostic approach. The presented results are very promising, however a bit preliminary. Thus, I would to bring some questions to the authors.

1.    Lines 49-61: The paragraph needs some references, mainly concerning the EPR effect differences in rodents and humans.

2.    The HER-2+ breast tumors must be more explored in the introduction, to fundament the author’s aim. 

3.    Lines 124-125: The anesthetic drug could be shown by the chemical name.

4.    The encapsulation efficiency must be added to the results clearly. I can not find these results even in the cited work (https://doi.org/10.3390/pharmaceutics15010052).

5.    In Figure 4 the merged image could be shown in a dark field either. 

6.    Why were not more images obtained with blank nanoparticles (loaded with fluorescent dye)?

7.    How many mice were used per group?

8.    What was the doxorubicin dose injected? 

9.    Preliminary data on toxicity (body weight) must be provided. 

Author Response

We thank the reviewer for these valuable comments, which are addressed below in point-by-point mode.

Comment 1

Lines 49-61: The paragraph needs some references, mainly concerning the EPR effect differences in rodents and humans.

Reply 1

We thank the reviewer for the valuable comment. The paragraph is now supported with the relevant references as follows:

“Considering the anticancer properties of medical nanostructures, it is worth noting that they reach the tumor due to the effect of increased permeability and retention of tumor vessels (EPR effect), which allows the accumulation of a significant proportion of injected nanoparticles in the tumor site. However, recent studies revealed that the EPR effect is not the main driving force in the accumulation of nanoparticles in human tumors [7–9]. The EPR effect is much more pronounced in rodents with rapidly developing tumors that do not have time to form a normal vascular network compared to large solid tumors in humans [10]. In this regard, there is an urgent need to develop drugs that differ in the mechanism of action from traditional drugs based on nanoparticles, such as the liposomal form of doxorubicin (Myocet and Caelyx) [11,12]. Specifically, targeted drug delivery systems (DDS) have improved characteristics in terms of accumulation in the tumor, but their use to this day remains controversial in biomedicine. In particular, to date, there are no targeted nanomedications approved by the FDA for the treatment of diseases [13]”.

Comment 2

The HER-2+ breast tumors must be more explored in the introduction, to fundament the author’s aim.

Reply 2

The relevant description was added to the introduction as follows:

“Targeted drug delivery is an important tool in the treatment of HER2-positive tumors, which are known for their aggressive nature. The human epidermal growth factor receptor 2 (HER2 receptor) is an important target for drug delivery in the treatment of HER2-positive tumors [14]. This receptor is found on the surface of certain types of cancer cells, and when activated by drugs, it can help to stop the growth and spread of these tumors. Drugs that target the HER2 receptor can be used alone or in combination with other treatments to improve patient outcomes. Studies have shown that using drugs that specifically target the HER2 receptor can help to reduce tumor size and slow or stop tumor growth [15,16]. Additionally, targeting this receptor has been shown to increase survival rates and reduce recurrence rates in patients with HER2-positive tumors [17]”.

Comment 3

Lines 124-125: The anesthetic drug could be shown by the chemical name.

Reply 3

The issue was corrected as follows:

“The animals were anesthetized with a mixture of tiletamine/zolazepam/xylazine at a dose of 20/20/2 mg/kg (Zoletil (Virbac, Carros, France) and Rometar (Bioveta, Ivanovice na Hané, Czech Republic))”.

Comment 4

The encapsulation efficiency must be added to the results clearly. I can not find these results even in the cited work (https://doi.org/10.3390/pharmaceutics15010052).

Reply 4

These results are presented in the previously published paper as a separate figure with a clear description in the main text. The encapsulation results are now cited in the submitted manuscript shortly as “We previously shown that the incorporation of doxorubicin incorporation was found to be 0.9 nmol doxorubicin per 1 mg of particles”, since we do not support the idea to duplicate previously published results (doi:10.3390/pharmaceutics15010052):

“Nanoparticles were dissolved in DMSO and then fluorescence was measured using a fluorescence calibration curve for doxorubicin samples in the same solutions (Figure 6a). The measurement of the fluorescence of the samples showed that doxorubicin incorporation was 0.9 nmol doxorubicin per 1 mg of nanoparticles,

Figure 6. Characterization of PLGA nanoparticles. (a) Calibration curve for the measurement of doxorubicin*HCl incorporation into PLGA nanoparticles obtained with fluorescence spectroscopy (excitation 480 nm, emission 590 nm). The red dot indicates the measured doxorubicin concentration in a 1 g/L sample of PLGA particles”.

Comment 5

In Figure 4 the merged image could be shown in a dark field either.

Reply 5

The merged images without bright field (dark) are added to the Figure 4.

Comment 6

Why were not more images obtained with blank nanoparticles (loaded with fluorescent dye)?

Reply 6

The blank nanoparticles, the nanoparticles loaded with Nile Blue, the nanoparticles loaded with doxorubicin, the blank nanoparticles conjugated to barnase, and the combined nanoparticles with Nile Blue, doxorubicin and barnase were thoroughly characterized by us previously and the results were already published. The separate paper was devoted to this characterization and in the submitted work we only confirmed the colloidal stability of nanoparticles and provide SEM images of the current batch of nanoparticles used for in vivo study.

The published results are shown in Figure 3 of our previous paper (doi:10.3390/pharmaceutics15010052) as follows:

Figure 3. Characterization of PLGA nanoparticles. (a) Scanning electron microscopy analysis of pristine polymer nanoparticles (PLGA), particles loaded with Nile Blue only (PLGA-Nile Blue), particles loaded with doxorubicin only (PLGA-Dox), particles conjugated with barnase (PLGA-Bn), particles loaded with doxorubicin, Nile Blue and conjugated with barnase (PLGA-DOX-Nile Blue-Bn). (b) The physical size distribution of nanoparticles loaded with Nile Blue and doxorubicin is obtained by image processing. (c) Hydrodynamic sizes, polydispersity indices, and ζ-potentials of pristine PLGA nanoparticles, particles loaded with doxorubicin only (PLGA-Dox), particles loaded with Nile Blue only (PLGA-Nile Blue), particles loaded with doxorubicin and Nile Blue (PLGA-Dox-Nile Blue), particles loaded with doxorubicin and Nile Blue (PLGA-Dox-Nile Blue) and stored for 1.5 years, particles loaded with doxorubicin and Nile Blue and conjugated with barnase (PLGA-Dox-Nile Blue-Bn)”.

Comment 7

How many mice were used per group?

Reply 7

The number of mice was n = 3, all the treated mice are presented in Figure 5. The number of mice in groups is indicated in the manuscript as follows: “EMT-HER2 tumor growth dynamics under the treatment with one-step DDS, PLGA*DARPin9_29 particles, or two-step DDS, Bs-DARPin9_29 & PLGA-Bn (n = 3 mice for each group)”.

Comment 8

What was the doxorubicin dose injected?

Reply 8

The issue is highlighted in the text as follows: “According to the treatment scheme, mice received totally 0.9 nmol of doxorubicin (by two sequential injections of 500 µg of nanoparticles equalto total of 1 mg of nanoparticles)”.

Comment 9

Preliminary data on toxicity (body weight) must be provided.

Reply 9

The toxicity data and biosafety tests were provided in the manuscript (Fig.7-8), namely  bodyweight is presented on Figure 8a.

Reviewer 2 Report

This study was well-established and demonstrated a nanoparticle strategy for tumor treatment. However, it is necessary to briefly explain the results of previous studies in the text. In this paper, the chemical structure of the polymer used in the manufacture of polymer nanocapsules and the chemical synthesis method and the process should be presented and described. If this part is supplemented, it can be published in this journal.

1.     Methods for fabrication and identification of PLGA nanoparticles should be added to sections 2.1 and 3.1.

2.     Direct results that can prove that each nanoparticle manufactured for One step DDS and Two step DDS is well prepared as described should be presented.

3.     Please add a description of the size, surface charge, PDI, and drug loading efficiency of PLGA nanoparticles.

Author Response

We thank the reviewer for these valuable comments, which are addressed below in point-by-point mode.

Comment 1

Methods for fabrication and identification of PLGA nanoparticles should be added to sections 2.1 and 3.1.

Reply 1

We thank the reviewer for this comment. The section 3.2  was already devoted to the synthesis and characterization of PLGA particles. However, to enhance the readability of the manuscript, we included the short description of the synthesis method in Section 3.1 as follows: “The synthesis of PLGA nanoparticles is based on double “water-oil-water” micro-emulsion technique (Figure 2a). Doxorubicin is incorporated into nanoparticle structure through the first “water” phase, while Nile Blue is incorporate into the “oil” phase [27]”. As for section 2.1 (Methods) we suppose not to add  the details of this synthesis, since it is already published in detail in (doi:10.3390/pharmaceutics15010052).

Comment 2

Direct results that can prove that each nanoparticle manufactured for One step DDS and Two step DDS is well prepared as described should be presented.

Reply 2

Indeed, this coment is very useful, however, the separate manuscript was devoted to this characterization (please, see doi:10.3390/pharmaceutics15010052).

Comment 3

Please add a description of the size, surface charge, PDI, and drug loading efficiency of PLGA nanoparticles.

Reply 3

The parameters of PLGA nanoparticles (size, surface charge, PDI, and drug loading efficiency) was already presented in as follows (please, see doi:10.3390/pharmaceutics15010052):

Figure 3. Characterization of PLGA nanoparticles. (a) Scanning electron microscopy analysis of pristine polymer nanoparticles (PLGA), particles loaded with Nile Blue only (PLGA-Nile Blue), particles loaded with doxorubicin only (PLGA-Dox), particles conjugated with barnase (PLGA-Bn), particles loaded with doxorubicin, Nile Blue and conjugated with barnase (PLGA-DOX-Nile Blue-Bn). (b) The physical size distribution of nanoparticles loaded with Nile Blue and doxorubicin is obtained by image processing. (c) Hydrodynamic sizes, polydispersity indices, and ζ-potentials of pristine PLGA nanoparticles, particles loaded with doxorubicin only (PLGA-Dox), particles loaded with Nile Blue only (PLGA-Nile Blue), particles loaded with doxorubicin and Nile Blue (PLGA-Dox-Nile Blue), particles loaded with doxorubicin and Nile Blue (PLGA-Dox-Nile Blue) and stored for 1.5 years, particles loaded with doxorubicin and Nile Blue and conjugated with barnase (PLGA-Dox-Nile Blue-Bn).

Nanoparticles were dissolved in DMSO and then fluorescence was measured using a fluorescence calibration curve for doxorubicin samples in the same solutions (Figure 6a). The measurement of the fluorescence of the samples showed that doxorubicin incorporation was 0.9 nmol doxorubicin per 1 mg of nanoparticles,

Figure 6. Characterization of PLGA nanoparticles. (a) Calibration curve for the measurement of doxorubicin*HCl incorporation into PLGA nanoparticles obtained with fluorescence spectroscopy (excitation 480 nm, emission 590 nm). The red dot indicates the measured doxorubicin concentration in a 1 g/L sample of PLGA particles”.

Reviewer 3 Report

First of all, congratulations for the good work done. It is a very complete study, with great projection for the future and is very well structured.

I would like to resolve some doubts and make some comments. Thanks in advance.

- In section 3.4. For in vivo studies, how many animals were used for each study group? Based on what were days 8 and 10 of treatment administration selected?

- On line 378, the subsection is wrongly numbered.

- And on line 411, does it correspond to section number 4 or 5?

- The References section, adapt it to the corresponding font.

Author Response

We thank the reviewer for these valuable comments, which are addressed below in point-by-point mode.

Comment 1

In section 3.4. For in vivo studies, how many animals were used for each study group? Based on what were days 8 and 10 of treatment administration selected?

Reply 1

We thank the reviewer for the valuable comment, indeed, the number of animals were missed and now presented as “EMT-HER2 tumor growth dynamics under the treatment with one-step DDS, PLGA*DARPin9_29 particles, or two-step DDS, Bs-DARPin9_29 & PLGA-Bn (n = 3 mice for each group)”.

Days 8 and 10 were selected for the nanoparticle injection based on tumor growth dynamic study: when the tumor volume reached 200-300 mm3 thus corresponding to 0.1-0.15 % of mouse body weight. These tumors are big enough with promounced vasсularization and can serve as a model of big human solid tumors.

Comment 2

On line 378, the subsection is wrongly numbered.

Reply 2

Corrected.

Comment 3

And on line 411, does it correspond to section number 4 or 5?

Reply 3

The section number is 4, the issue was corrected.

Comment 4

The References section, adapt it to the corresponding font.

Reply 4

Corrected.

Round 2

Reviewer 1 Report

Dear authors,

Thank you for all the modifications to the manuscript. However, I still have a query about the images. I would like to know about the tumor-bearing mice images. Why were more images of animals not performed using only the blank formulation? More time points images could be provided to support the tumor volume results. 

Author Response

The main goal of this study was to compare the efficacy of one-step and two-step drug delivery strategies (DDS). All in vitro tests were carried out and described in our previously published work (doi 10.3390/pharmaceutics15010052). The data presented in Figure 10 of (doi 10.3390/pharmaceutics15010052) indicate that nanoparticles not modified with proteins do not have cytotoxic activity; therefore, it is not ethical to conduct animal experiments to test these ineffective nanoparticles.

Considering the number of images, it is inappropriate to evaluate tumor volume results based on fluorescence measurements. Fluorescence detection in vivo is not quantitative due to differences in light absorbance and dissipation at different depths of tumor tissue; thus, it is incorrect to evaluate the tumor volume during all the monitoring time. The caliper measurements were performed instead and presented in Figure 6.

Moreover, since nanoparticles undergo degradation in vivo, mice should receive additional doses of therapeutic nanoparticles in order to localize the tumor site on days from 10 to 20. However, we do not deem this reasonable because the tumor growth inhibition was equal to 94.9% for two-step DDS compared with 68.4% for one-step DDS: both indices demonstrate quite effective treatment and no additional doses are required.

Again, we believe that it is unethical to use two more groups of mice separately just for examining fluorescent tumor volume by sequential injection of more than two doses of nanoparticles (three or more). All the necessary data have already been obtained: both therapeutic and diagnostic efficacy of the particles have been shown. Further studies on degradation and quantitative accumulation of such structures are being carried out by us now and will be published in future works.

Round 3

Reviewer 1 Report

All my questions were answered by the authors. 

I recommend the publication after a careful revision of English.